# Using Classify-While-Scan (CWS) Technology to Enhance Unmanned Air Traffic Management (UTM)

**Jiangkun Gong** [1] [ID]**, Deren Li** [1]**, Jun Yan** [1,*]**, Huiping Hu** [2] **and Deyong Kong** [3]

1    State Key Laboratory of Information Engineering in Surveying, Mapping and Remote Sensing,
     Wuhan University, Wuhan 430072, China
2    Wuhan Geomatics Institute, Wuhan 430022, China
3    School of Information Engineering, Hubei University of Economics, Wuhan 430205, China
*    Correspondence: yanjun_pla@whu.edu.cn; Tel.: +86-027-68778527

**Abstract:** Drone detection radar systems have been verified for supporting unmanned air traffic management (UTM). Here, we propose the concept of classify while scan (CWS) technology to improve the detection performance of drone detection radar systems and then to enhance UTM application. The CWS recognizes the radar data of each radar cell in the radar beam using advanced automatic target recognition (ATR) algorithm and then integrates the recognized results into the tracking unit to obtain the real-time situational awareness results of the whole surveillance area. Real X-band radar data collected in a coastal environment demonstrate significant advancement in a powerful situational awareness scenario in which birds were chasing a ship to feed on fish. CWS technology turns a drone detection radar into a sense-and-alert planform that revolutionizes UTM systems by reducing the Detection Response Time (DRT) in the detection unit.

**Keywords:** automatic target recognition (ATR); classify while scan (CWS); drone detection radar; detection response time (DRT); unmanned air traffic management (UTM)

## 1. Introduction

Within the past few years, the number of unmanned aircraft systems (UAS), also often referred to as drones, has increased rapidly. They play essential roles in various tasks, including both civil applications and military applications. Drones are often used for reconnaissance, communication, and even attack missions by military clients. In 2020, drones revealed a new war era using the drone and anti-drone war after they achieved brilliant results in the Nagorno-Karabakh conflict [1,2], such as STM Kargu, Bayraktar TB2, IAI Harop, Orbiter 1K, etc. In addition to the military applications, drones, especially small drones, are popular in many civil projects [3], including entertainment photography, sports recording, infrastructure monitoring, precision agriculture, package delivery, rescue operations, remote mapping, and more. It seems that the era of drones will be the near-term future.

The flush drone applications require urgent counter UAS systems (C-UAS). C-UAS are generally designed to eliminate drone targets, and this includes three steps: (1) detection, (2) tracking and identification, and (3) effector [4,5]. The detection unit generally uses radar systems to detect the possible threat, and then operators identify the threat using EO/IR systems; finally, they utilize the effector unit to defeat the threat. Unlike military applications, some civil applications require that a C-UAS solution replace the defeater unit with a management unit. The typical application is airspace management at airports. An effective unmanned aircraft traffic management (UTM) system has been investigated for managing drones flying at low altitudes around airports [3,6].

The concept of UTM systems originates from the current air traffic management (ATM) system supporting flight operations. Most of the detection workload is taken by ground-based radar systems, such as airport traffic control (ATC) radar [7,8]. The primary

identification work is supported by mandatory ACK systems such as the ADS-B system and the TCAS. One of the most challenging problems in designing and applying UTM is that there are neither associated EO/IR systems nor mandatory ADS-B sensors [9]. In this case, the radar system should take the task of identification. If a radar system can automatically detect radar echoes from drones and identify them from other clutters, it can considerably improve the performance of the UTM system. In other words, the drone detection radar needs to be equipped with automatic target recognition (ATR) module.

A radar ATR function is defined as recognizing targets mainly based on radar signals. Primitive target recognition was performed by using the audible representation of the received echoes, and then a trained operator deciphered the information in the sound. Later, radar engineers and scholars conducted design algorithms to do the work automatically. After decades, several schools have recognized radar features that succeed in ATR applications in some cases. Note that the traditional ATR solution contains two procedures, including feature extraction and pattern recognition. Here, we use the feature to classify the interpretations of ATR. The first one is high-range resolution profile (HRRP) technology [10]. Radar transmits ultrawide-band signals and obtains the target profiles in the range direction. The HRRP is a mapping of the shape of the target, and it is processed with a template matching approach in the dataset to obtain the target class. The second one is the micro-Doppler [11]. Micro-Doppler is thought to be the additional Doppler component produced by the micro motions on the target, such as the rotating movements of helicopters' blades, flapping birds' wings, and more. The embedded kinematic/structural information of the target can be measured from the micro-Doppler and then used for registering the target [11]. In addition, the third method could sometimes use tracking information such as speed and trajectory for identifying targets. Although machine learning is popular in recent years [9,12,13], it is a black-box algorithm to process the units of feature extraction and pattern recognition. Since machine learning in ATR applications is an approach over a feature, it is not discussed here. Generally, a machine learning method can process either HRRP data, micro-Doppler data, or tracking data to recognize the target from different background cluttered environments. In addition, all these radar data can be presented via radar images or signals.

Currently, the discussion of applying ATR to drone detection projects or other projects underestimates the value of an ATR algorithm. This paper proposes the higher value of ATR functions, which is the situational awareness (SA) ability using the patented classify while scan (CWS) technology. Section 2 presents the basic introduction of CWS technology and our radar platform. Section 3 describes and analyzes some experimental results, and the application is discussed in Section 4. Finally, our conclusion is presented in Section 5.

## 2. Materials and Methods

### 2.1. Design Principle

Generally, most ATR functions focus on the "point" information. The "point" characteristics are in both time and space dimensions. Spatially, they often recognize radar signals in a single radar resolution cell, and they also focus on current radar echoes over past and future radar signals temporally. They neither use tracking data of the target nor connect the radar signals in the radar resolution cells around the target. To the best of our knowledge, the track while scanning (TWS) could be probably the only technology that uses the time information of the target.

TWS is a mode of radar operation in which the radar scans the surveillance area while the acquired targets are tracked [14]. The significance of TWS is that the radar provides an overall view of the surveillance area and helps maintain better situation awareness (SA). Radar systems equipped with the TWS function sometimes also claim that they are 4D radar, in which time labels of the target are also regarded as the fourth dimension together with traditional 3D information (i.e., azimuth, elevation angle, and slant range). However, it seems that 4D radar is just a sales-promoting buzzword that has nothing to do with the

fourth dimension in physics. In our opinion, the real 4D radar should include the target attribute given by an ATR function.

Integrating ATR and TWS is the technology of classify-while-scan (CWS). As shown in Figure 1, CWS processes the raw data in one radar resolution cell and obtains the object's ID. The ID of the target can be either used for recording, displaying on the radar screen, or assisting the tracking unit. With the ID of the target, the tracking unit could easily connect the same ID between the contiguous tracking data. This process can be called track-after-identify (TAI). Then, the CWS function processes radar data in every radar cell and outputs the targets along with the ranged cells in the radar beam. Consequently, the radar beam continues to scan the area and capture the traces of the active targets, and then the whole scenario is presented in a radar display following the scan of the radar beam. Figure 2 demonstrates the basic flowchart of the CWS function.

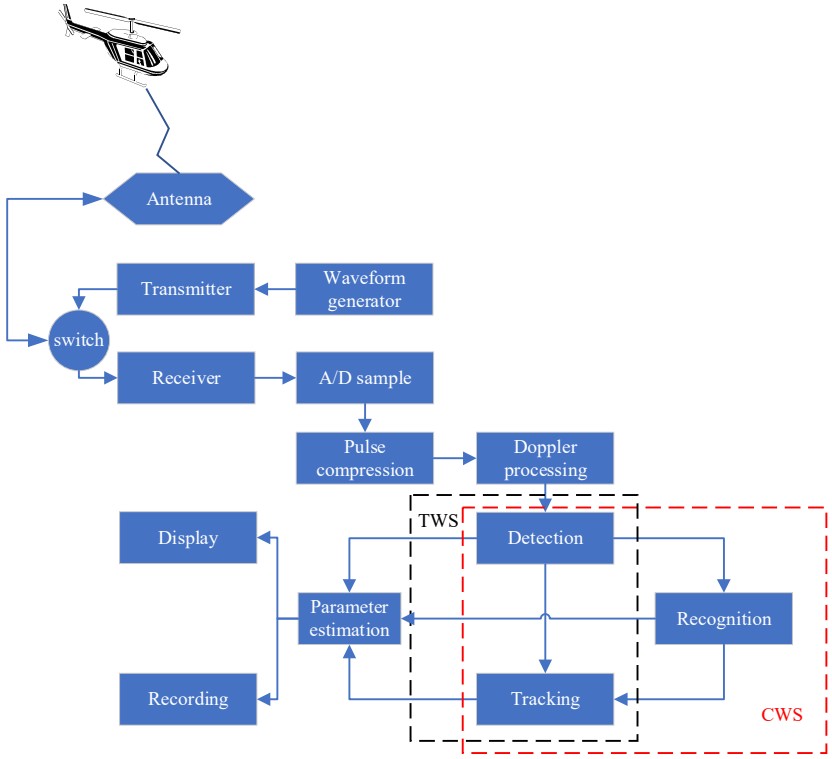

**Figure 1.** The block diagram of radar equipped with CWS and TWS technology.

The specific algorithms in CWS technology can differ with different methods. For example, the detection methods can include an algorithm based on the signal-to-noise ratio (SNR), the moving target indication (MTI), the algorithm extracting the signal-to-clutter ratio (SCR), and more. ATR algorithms can be diverse, such as HRRP, micro-Doppler, and time-frequency analysis. The only difference exists in the tracking algorithm. Traditional tracking algorithms mainly depend on the correlation of Doppler and trajectory. However, CWS provides ID labels of objects in the radar beam, and then tracking algorithms can use ID labels to improve the correlation process and enhance the tracking accuracy. As shown in Figure 2, our CWS processing algorithm contains several steps, including:

(1)   Assume there are N radar range cells in one radar beam;
(2)   Extract the radar signals of objects using our SCR detector [15,16];
(3)   Recognize the radar echoes of objects using radar signal signatures;
(4)   Repeatthe above detection & recognition algorithms and obtain all the IDs in each radar range cell;
(5)   The radar tracking unit uses the recognized results (i.e, IDs of targets) to track targets.

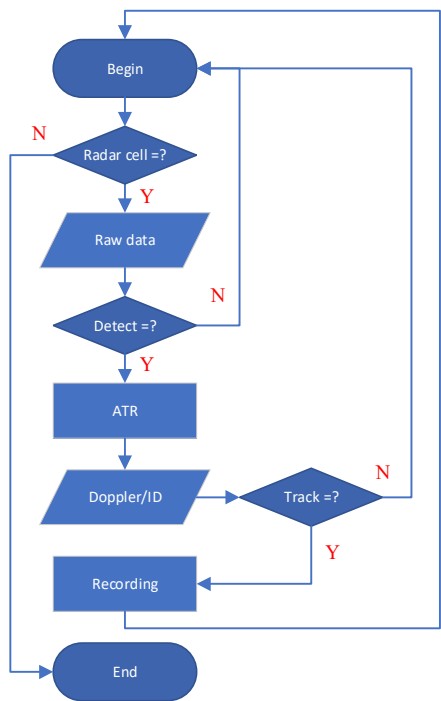

**Figure 2.** The basic flowchart of CWS technology.

## 2.2. Test Conditions

Here, we demonstrate a drone detection radar system and the detection results using CWS technology. We applied the CWS technology to a new coastal surveillance radar. The pulse-Doppler radar works in the X-band. It is a narrow-band radar with a range resolution of approximately 12 m. The radar is equipped with an active, electronically scanned, phased-array antenna and deployed on a rotating table to achieve 360-degree coverage in the azimuth scan. Moreover, it uses digital beam-forming (DBF) technology to obtain multiple radar beams in the pitch direction every time. The flexibly configured rotating speed of the rotating table is between 2 s and 20 s. The detection response time (DRT) is approximately 30 ms, representing that the lag between an echo return, detection, and eventual display is only 30 ms. It is capable of tracking 1000 targets simultaneously. It can recognize different targets, including birds, drones, vehicles, ships, people, helicopters, and others, and then it presents detection results with graphic icons, which label the recognition results, along with the rotating motion of the scanning beam. The numbers around the icons are tracking numbers.

The test was conducted in a coastal area with a cluttered sea environment. The test area is located on the Yellow Sea coast of Qidong, China. The radar is set on the roof of a 12-m tall building, and the sea was scanned horizontally. The initial goal of this project was to develop a prototype drone detection radar. During the project, an infrared sensor and an optical camera were deployed to support the project and used to confirm the recognition results of the ATR function. This project continued for several months in 2020, and some data were extracted in this paper. The sea scale during the test was about Degree 3~5. The height of the wave was 0.5–1.25 m with Degree 3 and 2.50–4.00 m with Degree 5.

We collected radar signals of different types of drones, including a quad-rotor drone, a fixed-wing drone, and a hybrid Vertical Take-off and Landing (VTOL) fixed-wing drone. They were cooperative targets. Some of their parameters are listed in Table 1. Albatross 1 is a homemade fixed-wing drone with only one pusher blade, DJI Phantom 4 is a famous quad-rotor drone with four lifting blades, and TX25A is a large hybrid VTOL fixed-wing drone with one pusher blade and four lifting blades. Photos of these drones are shown in Figure 3. If we refer to NATO's category of drones [4], Albatross 1 is a Microdrone, DJI

Phantom 4 is a Mini-drone, and TX25A is a Small-drone. In addition, some local fishing ships and birds were also our test targets, and we also collected their radar data.

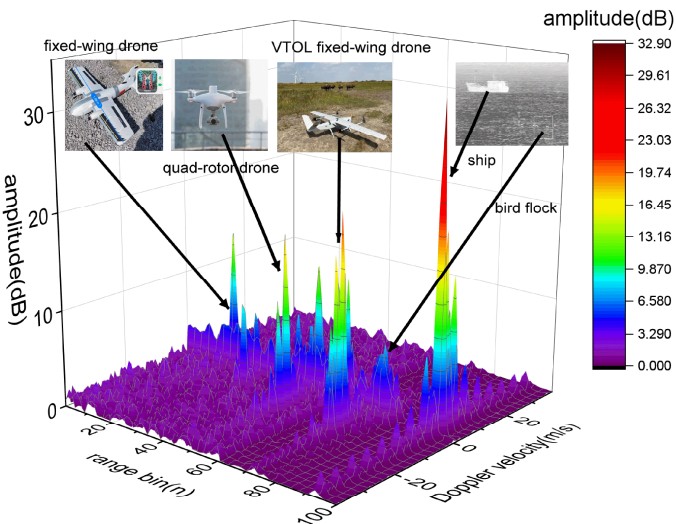

**Figure 3.** The range-Doppler image of radar data in one radar beam containing different objects.

**Table 1.** Parameters of different drones.

| Drone Type | Hybrid VTOL Fixed-Wing | Fixed-Wing | Multi-Rotor |
|---|---|---|---|
| Model | TX25A | Albatross 1 | Phantom 4 |
| Manufacturer | Harryskydream Inc. | Homemade | DJI Inc. |
| Flight weight | 26 kg | 0.3 kg | 1.38 kg |
| Body size | 197 cm | 80 cm | 40 cm |
| Wing span | 360 cm | 108 cm | 40 cm |
| Cruise speed | 25 m/s | 10 m/s | 15 m/s |
| Rotor number | 5 | 2 | 4 |
| Blade length | 30 cm | 10 cm | 20 cm |
| Aero-frame materials | FRP (Fibre reinforces plastic) | EPP (Expanded polypropylene) | PC (Polycarbonate) |

## 3. Results

It required several steps from an echo return into the eventual display on the radar display. Figures 3–5 show some examples of the whole procedure, and Figure 6 demonstrates a real radar screenshot using CWS technology. Figure 3 shows the range-Doppler images containing several specific objects in one simulated radar beam. Each radar range sector had twenty bins, where the target was in the center of the range sector. The range resolution of a radar bin is 12 m, and the range of a range sector is 240 m. To demonstrate the detection results, we manually jointed five sectors from different radar beams into the detection result in Figure 3. Most clutters were approximately 0 Hz. Since the detection ranges of those targets were different, the amplitudes of the signals were incomparable. Nevertheless, the Doppler signals in each spectrum still differ from each other. We use the SCR detector to extract radar signals of targets in cluttered backgrounds [15]. The SCR detector performs superior to the SNR detector, reducing missed and false alarms when detecting and tracking drones.

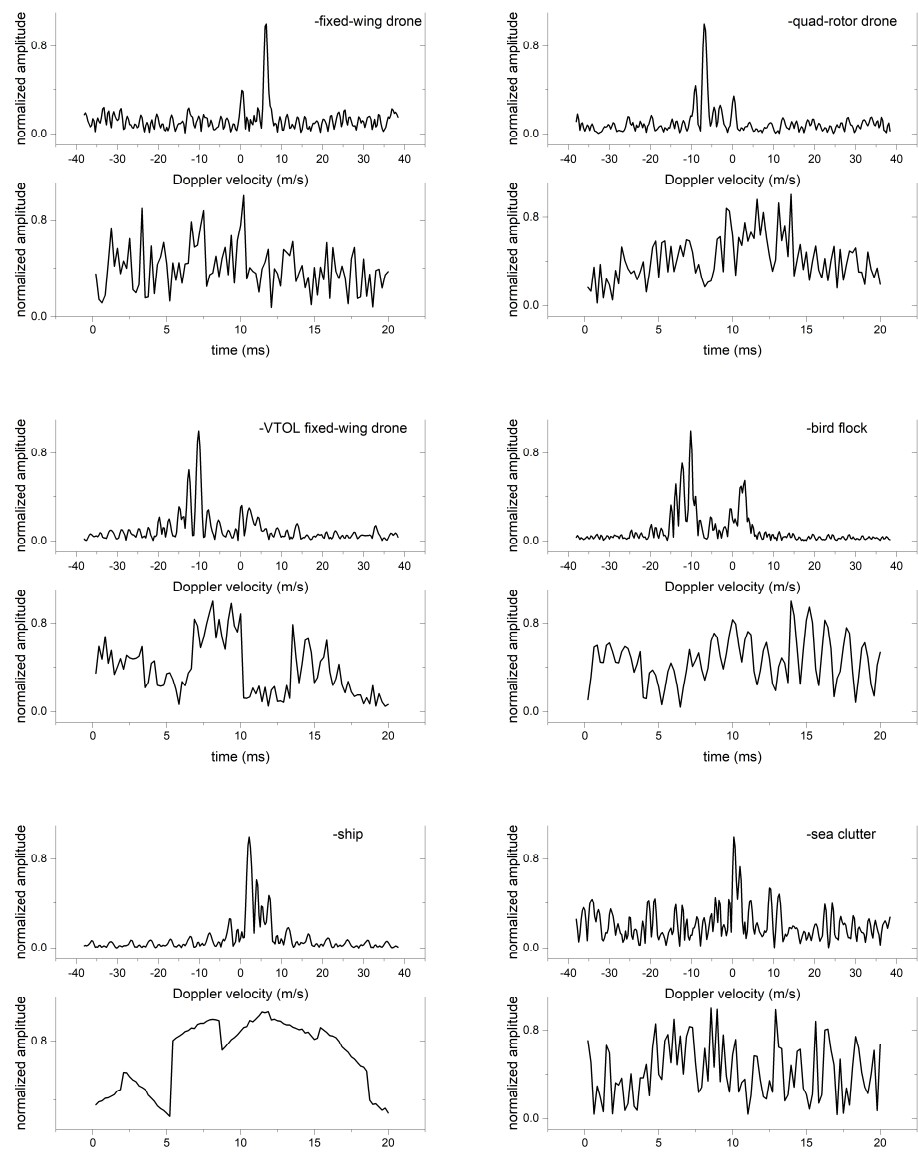

**Figure 4.** Raw time & frequency data of different objects.

The time-frequency characteristics of different objects provide useful signatures for the ATR module. Figure 4 abstracts the raw radar of objects in Figure 5 and plots the time-frequency distributions of each object. The spectra were obtained using conventional Fourier transforms. We normalized the signal amplitudes to remove the interferences posed by the detection ranges. The radar dwell time in the time domain was 20 ms. There were no patterns in the time domains. In addition to the body Doppler, the micro-Doppler also appeared in the spectra, and they seem to have patterns. Then, a patented ATR algorithm is used to analyze the time-frequency characteristics of radar signals from different objects and obtain the recognized results of the targets.

The radar tracking units used the recognized results in each radar beam to enhance the tracking data of every target. Figure 5 shows the tracking data of our targets in Figures 3 and 4. These data were real data collected on different test dates. The height information was calculated using DBF technology; therefore, the accuracy was limited. Nevertheless, the trace of each target still described the moving kinematics of each target. Generally, the mean Doppler velocity values of the fixed-wing drone, quad-rotor drone, VTOL fixed-wing drone, bird flock, and ship were approximately 7.34 m/s, 6.76 m/s, 15.22 m/s, 12.75 m/s, and 3.36 m/s, respectively. The ship has the slowest moving speed,

while the VTOL drone has the fastest moving velocity. In addition, the measured height numbers of those targets were approximately 670.07 m, 272.68 m, 100.05 m, 20.81 m, and 0 m, respectively. These measured characteristics were following the natural flight parameters of these objects.

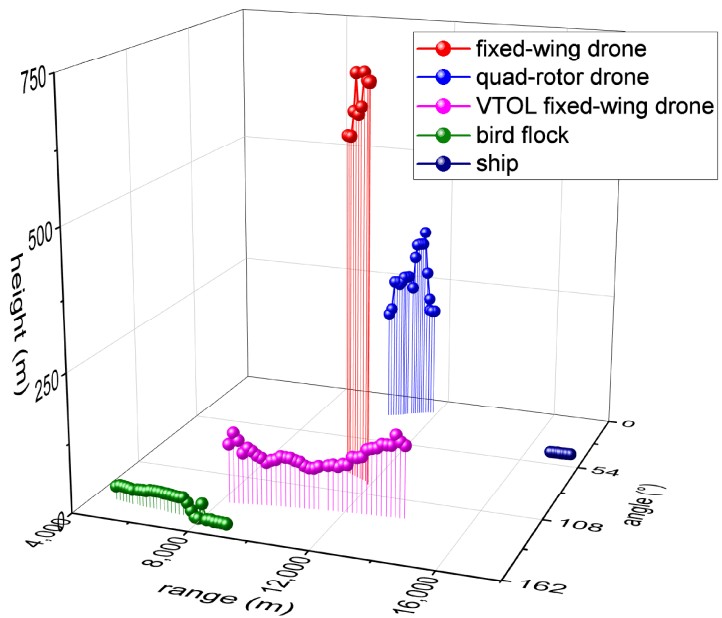

**Figure 5.** The tracking data of different moving targets.

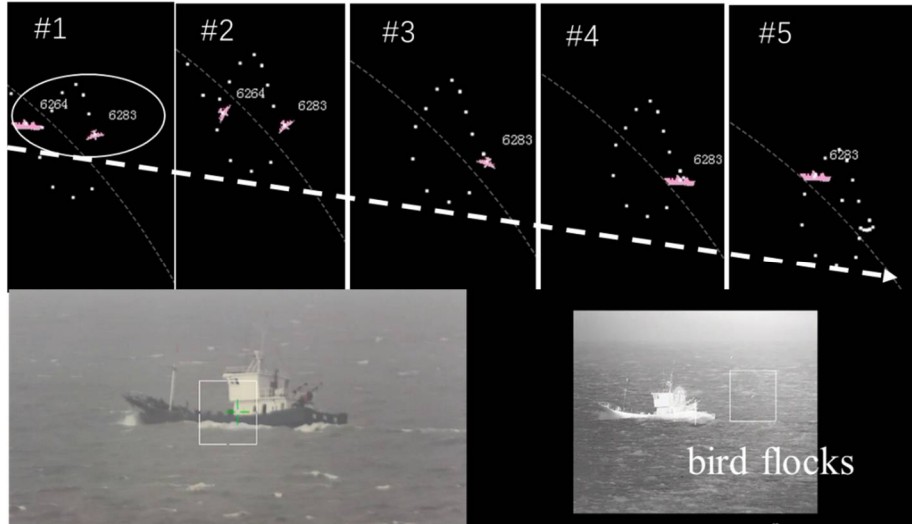

**Figure 6.** A real scenario sensed by situation awareness using the CWS technology, where the birds were chasing the ship to feed on fish.

## 4. Discussion

The most significant benefit provided by CWS technology is situational awareness (SA). Figure 6 provides a typical example of an application of situational awareness where birds chased ships to pick up fish awakened by the ship's propellers. The recognized results were marked using graphic icons, including ships, helicopters, drones, and birds. The thin white dotted lines are the tracking traces of objects. The numbers around the icons were the tracking numbers of objects. CWS enhanced the radar to become a WYSIWYG (What You See Is What You Get) system or a real-time sense-and-alert system. Moreover, the whole scenario in the surveillance area was captured and updated with new data. Note

the tracking birds of No. 6264 were flying around the ship of No. 6283; thus, radar echoes from birds were interfering with the detection of the ship. When they appeared in the same radar cell, the technology recognized that these data were sometimes birds while the others were the ship. The birds were flying around the ship because they habitually chase ships in the sea to feed on the fish, which were awakened by the rotating propellers of the ship. In other words, SA provided by CWS described a habit of sea birds.

UTM is a safety system that helps ensure the newest entrant, a variety of drones, into the skies does not collide with buildings, larger aircraft, or one another. UTM is different from the ATM functioned by the airport agencies in some respects. First, UTM cannot use the airport traffic control (ATC) radar systems used by the current ATM system. ATC radar systems are primarily designed to detect and track large, fast-moving aircraft flying in high airspace. Compared to large aircraft, drones generally have small RCS values and slow-moving speeds. In particular, the ATC radar systems were reported to turn to identify and retain targets that move consistently, remain visible from sweep to sweep, and have a ground speed of at least 15.432 m/s, and, as a result, they failed to detect the flock of Canadian geese in the aircraft accident of flight 1549 on 15 January 2009 [17]. The flight characteristics of drones make them potentially undetectable by traditional ATC radar systems. Second, drones may not be equipped with signal transmission sensors, such as RID broadcast receivers and ADS-B. This means that ground-based detection systems could be critical sensors to support the UTM system.

Similar to the ATC in the ATM system, a drone detection radar system is the key role of UTM. Nevertheless, not all drone detection radar systems are suitable. Here, we insist that a 4D radar must detect an object and obtain its 3D position and 1D attribution, provided by an ATR algorithm. Figure 7 demonstrates a typical application, where 4D radar can enhance the real-time situational awareness at a simulated airport. In this way, the drone detection radar can detect and track different types of drones (e.g., quad-rotor drone, VTOL drone, unmanned-helicopter, etc.) over other large aircraft or bird clutters and then support a UTM system. ATR turns the radar data of objects into meaningful IDs of targets. These knowledge-based IDs could also be used for other applications, such as human-machine distributed situational awareness [9]. SA may be a key in UTM applications.

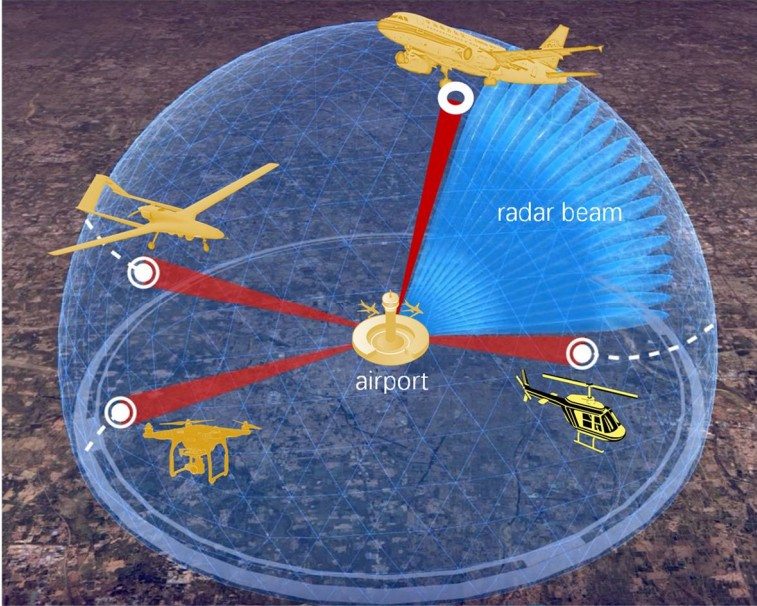

**Figure 7.** Using CWS to enhance UTM at an airport.

Generally, the performance of an ATR algorithm can be described in four tiers:

(1) Tier DETECTION: extracting radar signals of a target in background clutter, such as whether the radar echoes are from a target or a clutter;

(2) Tier CLASSIFICATION: classifying the class of the target, such as whether the radar echoes are from a drone or a bird;

(3) Tier IDENTIFICATION: identifying the type of the target in one class, such as whether the radar echoes are from a fixed-wing drone or a multi-rotor drone;

(4) Tier DESCRIPTION: describing the model of the target, such as whether the radar echoes are from a DJI Phantom 4 drone or a DJI S800 drone.

There are many types of aircraft at airports, so the ATR function should achieve the Tier IDENTIFICATION to support the UTM application (e.g., the example in Figure 7). As a higher tier of ATR function, CWS technology provides valuable situational awareness to support the UTM system. The performance of CWS technology must consider three factors, including the detection range, processing speed, and confidence factor. First, no matter what the algorithm of a CWS function is, it should be independent of the detection range of a target. Otherwise, the drone detection radar could not cover enough airspace at airports. Second, the processing speed determines the DRT. There is always a lag between an echo return and eventual display on the radar screen. The lag is sometimes called the DRT, which contains the time spent on communication and processing. Ideally, DRT should be at a level of milliseconds to fulfill a WYSIWYG system. Practically, the DRT should be shorter than the update interval between the current radar beam and the next radar beam. Last, but not least, the confidence factor determines the recognized result of CWS technology. Unlike the identification probability, the confidence factor describes the trust level of the recognized results calculated by the ATR algorithm of the CWS technology. Besides, the trajectory prediction is a key module in the ATM/UTM system [18–23], which can use the predicted trajectory to reduce the DRT in the tracking unit. In contrast, our CWS technology shortens the DRT in the detection unit. If the CWS technology is deployed in the radar system, it can provide the location of a target for the tracking unit using trajectory prediction algorithms, faster. In total, the shorter DRT and a higher confidence factor of a good CWS technology can result in a better UTM system.

## 5. Conclusions

There is a need for drone detection radar systems to support UTM systems at the airport. Here, we propose CWS technology to improve the detection performance of drone detection radar systems and enhance UTM systems. The CWS processes radar data of each radar cell in the radar beam using the ATR algorithm and then obtains all targets' recognized results. It reduces the DRT in the detection unit of the radar. Moreover, the recognized results can be used to track targets using the TAI algorithm in a surveillance area and then obtain real-time situational awareness (SA) results. The future work will investigate the performance of the TAI algorithm, and the performance of the radar situational awareness (SA) with the CWS and TAI algorithms. With situational awareness, drone detection radar has become a WYSIWYG, bringing revolutionary performance to UTM systems.

**Author Contributions:** Conceptualization, J.Y.; methodology, J.G.; software, D.K.; validation, J.Y.; formal analysis, J.G.; investigation, J.Y.; resources, D.L.; data curation, J.Y.; writing—original draft preparation, J.G.; writing—review and editing, H.H.; visualization, H.H.; supervision, J.Y.; project administration, D.L.; funding acquisition, D.K. All authors have read and agreed to the published version of the manuscript.

**Funding:** This research received some support by the Natural Science Foundation of Hubei Providence (General Program: 2021CFB309).

**Institutional Review Board Statement:** Not applicable.

**Informed Consent Statement:** Not applicable.

**Data Availability Statement:** Some of the data presented in this study may be available on request from the corresponding author. The data are not publicly available due to the internal restriction of the research group.

**Acknowledgments:** We appreciate both the testers during the collection of the data, and we also want to thank the authors whose photographs are reproduced in this study.

**Conflicts of Interest:** The authors declare that they have no conflict of interest.

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
