# Peer review of "Using Classify-While-Scan (CWS) Technology to Enhance Unmanned Air Traffic Management (UTM)"

_drones, doi:10.3390/drones6090224_

Round 1
Reviewer 1 Report
Please see the attachment.

Reviewer 2 Report
In general, the work was easy to read, however I have a few comments that show that the work should have been checked better before submitting.
1. VOTL is used four times, I believe this should be VTOL
2. Line 212 small is written twice
3. Fig 3. The numbers are poorly visible, the text on the axis is too small, some text is overlapping.
4. Fig 5. Exactly the same problems as in fig 3.
5. Fig 4. What is the red line going through the figures?
6. Fig 6. In the text, you talk about no.6283 and 6246. They are not visible. In general, the figure is difficult to see/understand.
Besides that, I have some other comments about the structure of the work.
7. The paper is not balanced, i.e. the results section is too long, the conclusions are too short, etc. My suggestion is to extend the materials and methods section so that it includes the text up until table 1. (this is still all materials and methods). Secondly, use subsections to better divide your sections. Lastly, the conclusions are too short.
8. On line 236 use enumeration, and one line for each item
Finally, some small comments
9. Around line 26 to 32, add some reference to the examples you are giving.
10. Line 70, are there other works that do use AI? If so, it might be interesting to reference them.
11. Line 24 "within the past few years, .... seem to skyrocket in one night". This is a weird sentence.
12. Does the laser have a brand name/ place where it can be bought? If so, please include a reference.
13. Is the raw radar data you gathered publicly available, this might be of great interest for other researchers.
Reviewer 3 Report
Authors propose to classify while scan (CWS) technology to improve the detection performance of drone detection radar systems and to enhance UTM applications.
Overall, the idea is clear and the technical contribution has some merit. However, this reviewer is very disappointed of the writing quality. The whole paper should undergo extensive english proofreading.
The structure of this 4-sections paper is also questionable. The reader cannot see the position of this paper compared to what has been already done. A related work section is highly required. The only good and detailed section is the discussion section. Please also show the organization of the paper at the end of the abstract.
Please extend a bit more your section titles and make them more representative.
The different parts of the paper should be extended with more details. The paper in its current form seem to be more a conference paper rather than a mature journal contribution.
What are the future directions of this work? Please include them to the conctusion section.
Round 2
Reviewer 1 Report
The revised manuscript has addressed my concerns and suggestions. Thus, I'd suggest accepting the paper as it is.
Reviewer 2 Report
All comments were addressed, no further changes are needed
Reviewer 3 Report
I think that the authors adequately addressed my comments. I see no reason to withhold acceptance.